# Functional selectivity of insulin receptor revealed by aptamer-trapped receptor structures

Junhong Kim [1,3], Na-Oh Yunn [2,3] ✉, Mangeun Park [2], Jihan Kim [1], Seongeun Park [2], Yoojoong Kim [1], Jeongeun Noh [2], Sung Ho Ryu [1] ✉ & Yunje Cho [1] ✉

Activation of insulin receptor (IR) initiates a cascade of conformational changes and autophosphorylation events. Herein, we determined three structures of IR trapped by aptamers using cryo-electron microscopy. The A62 agonist aptamer selectively activates metabolic signaling. In the absence of insulin, the two A62 aptamer agonists of IR adopt an insulin-accessible arrowhead conformation by mimicking site-1/site-2' insulin coordination. Insulin binding at one site triggers conformational changes in one protomer, but this movement is blocked in the other protomer by A62 at the opposite site. A62 binding captures two unique conformations of IR with a similar stalk arrangement, which underlie Tyr1150 mono-phosphorylation (m-pY1150) and selective activation for metabolic signaling. The A43 aptamer, a positive allosteric modulator, binds at the opposite side of the insulin-binding module, and stabilizes the single insulin-bound IR structure that brings two FnIII-3 regions into closer proximity for full activation. Our results suggest that spatial proximity of the two FnIII-3 ends is important for m-pY1150, but multi-phosphorylation of IR requires additional conformational rearrangement of intracellular domains mediated by coordination between extracellular and transmembrane domains.

The insulin receptor (IR) is a member of the receptor tyrosine kinase (RTK) family and a key regulator of metabolic homeostasis[1,2]. IR is a disulfide-linked $(\alpha\beta)_2$ dimer in which each protomer consists of α-and β-chains[3]. The extracellular α-chain is composed of a leucine-rich repeat (L1), a cysteine-rich region (CR), L2, fibronectin type III (FnIII-1), FnIII-2α, an insert domain (IDα), and the C-terminus of the α-chain (αCT). The β-chain contains extracellular IDβ, FnIII-2β and FnIII-3, transmembrane (TM) helix, intracellular juxtamembrane (JM), tyrosine kinase (TK), and C-terminal domains (Supplementary Fig. 1a). This architecture is also conserved in the insulin-like growth factor 1 (IGF-1) receptor[4]. Insulin binding to IR preferentially initiates *trans*-autophosphorylation at Y1146, Y1150, and Y1151 of the activation loop in the kinase domain, followed by phosphorylation of other Tyr residues:

Y953 and Y960 in the JM domain, and Y1316 and Y1322 in the C-terminal domain[5–9].

IR autophosphorylation on intracellular tyrosine residues stimulates two major signaling pathways: the AKT pathway initiated from insulin receptor substrate 1 (IRS1) and the mitogen-activated protein kinase (MAPK) pathway initiated from SHC-transforming protein 1 (SHC1)[10–12]. The metabolic effects of insulin, such as glucose uptake and glycogen and lipid synthesis, are mainly regulated by the AKT pathway. By contrast, the MAPK pathway is the primary mediator of the mitogenic effects of insulin, although the AKT pathway is also involved[12].

While insulin binding activates both major signaling pathways, the discovery of agonists with functional selectivity has revealed that IR activation by some ligands leads to selective downstream signaling

[1]Department of Life Sciences, Pohang University of Science and Technology (POSTECH), Pohang 37673, Republic of Korea. [2]Postech Biotech Center, Pohang University of Science and Technology (POSTECH), Pohang 37673, Republic of Korea. [3]These authors contributed equally: Junhong Kim, Na-Oh Yunn. ✉e-mail: beback13@postech.ac.kr; sungho@postech.ac.kr; yunje@postech.ac.kr

and cellular functions of IR in a manner distinct from that of insulin[13–15]. These ligands include antibodies, peptides, and nucleotides that selectively activate the AKT pathway and induce metabolic effects, but exert much weaker MAPK pathway signaling and mitogenic effects than insulin[13–17]. Moreover, some IR agonists also lead to site-specific mono-phosphorylation of Tyr1150 (m-pY1150) in the kinase domain of IR[17,18]. Thus, studies of these agonists of suggest that the early IR intermediates with site-specific Tyr phosphorylation in the activation event are responsible for the selective stimulation of the metabolic signaling[17].

Structural studies on IR bound to insulin revealed that the ligand can bind to up to four distinct sites, from which different IR activation models have been proposed[19]. In apo-IR, the extracellular domain of IR forms an inverted V-shaped dimer in which the two membrane-proximal FnIII-3 domains are separated by ~120 Å[20,21]. Single insulin binding to the primary site (named site-1) composed of L1 and αCT′ of the other pro-tomer induces an asymmetric Γ-shaped structure (a prime represents the second protomer)[22,23]. This conformational change leads to a large translocation of two FnIII-3 domains[21,23]. Similar conformational change of the two FnIII-3 domains was also observed in IGF1R complexed with IGF-1[24]. Structural change in the extracellular domain of IGF1R (or IR) induced by the ligand binding results in dimerization of the TM helices and autophosphorylation of the intracellular kinase domains[25]. As the insulin concentration increases, IR can interact with multiple insulins, which results in negative cooperativity of insulin binding (i.e., binding of a second insulin to IR weakens or releases the initially bound insulin)[26]. At saturated insulin concentration, four insulin molecules independently bind to two primary sites (site-1 and site-1′) and two secondary binding sites (site-2 and site-2′) at the FnIII-1 domain, which leads to a symmetric T-shaped structure[27–29]. However, in the apo-IR structure, the primary binding sites (L1 + αCT′) are closely located to the FnIII-2 domain, which prevents insulin access to the binding sites[20,21]. Thus, how insulin initially accesses IR remains puzzling, but the characterization of transient intermediate structures of IR is challenging due to their intrinsic instability. Furthermore, current models for IR activation cannot explain how the agonists selectively regulate receptor autophosphorylation and intracellular signaling in a manner distinct from insulin. The mechanism of action of selective agonists remains unclear due to a lack of agonist-bound structures.

Previously, we developed two DNA aptamers with modified bases that modulate autophosphorylation and downstream signaling of IR. IR-A62 (A62) is an aptamer agonist for IR that preferentially induces m-pY1150 in the kinase domain and selectively stimulates the AKT pathway and glucose uptake[18]. The IR-A43 aptamer (A43) is a positive allosteric modulator that enhances IR activation by stabilizing insulin binding[30]. Insulin and A43 bind to IR with mutual positive cooperativity, which potentiates IR autophosphorylation and downstream signaling.

In this work, we explore the selective activation mechanism of IR by trapping IR structures using A62 or A43 aptamers. We determine three distinct structures of the IR ectodomain bound to A62 or A43 aptamers, in the range of 3.62 to 4.27 Å resolution, in the absence or presence of insulin using cryo-electron microscopy (cryo-EM; Supplementary Table 1). The findings illuminate the conformational dynamics of IR activation potentially relevant for signaling, and provide a structural basis for designing functionally selective agonists for IR.

## Results
### Preparation of IR proteins complexed with aptamers
Full-length human IR (isoform A) was purified from the 293 F stable cell line overexpressing human IRs (Supplementary Fig. 1b–d). Because IR is a dimeric receptor linked by disulfide bonds, micelle formation of each TM domain can disturb the normal interaction between two TMs of an IR dimer[21]. Thus, IRs were complexed with aptamers in the presence or absence of insulin on the cell surface before cell lysis and

partial purification. Moreover, insulin-induced IR autophosphorylation triggers clathrin-mediated endocytosis of IR, which leads to insulin and IR degradation in late endosomes[31]. To improve the yield of complex formation, IR-aptamer complex formation (with and without insulin) on cells was carried out on the ice to block endocytosis and autophosphorylation of IR.

### Symmetric binding of two A62 aptamers induces the arrowhead IR conformation
We determined IR structures complexed with A62 alone (IR$_{2xA62}$), and with A62 and insulin (IR$_{A62+Ins}$), at an average resolution of 4.18 and 4.27/3.95 (global/local resolution) Å, respectively (Figs. 1a, 2a and Supplementary Figs. 1b–j, 2a–i). In the IR$_{2xA62}$ complex, the two A62 aptamers symmetrically bind to a receptor dimer (Fig. 1a–c). While all ectodomains are well-ordered, the TM and intracellular domains are not visible. The overall structure of the complex forms an arrowhead shape; the head consists of L1, CR, and L2 domains, and part of the FnIII-1 domain from both protomers, and the stalks are comprised of the two parallel FnIII-2 and FnIII-3 domains. The structure of the head domain is similar to that of the insulin-bound IR fragment[32] (5KQV, Supplementary Fig. 4a, b). The dimeric interface is formed by FnIII-1 and L2′ domains without extensive contacts (Fig. 1a, c). Compared with apo-IR, the structure of the entire protomer in the IR$_{2xA62}$ complex is not changed significantly (Supplementary Fig. 1k–m)[33]. Rigid body rotation of both protomers of apo-IR by 26.7° with respect to Cys524 (FnIII-1) resulted in the formation of the arrowhead IR conformation (Fig. 1d, e).

### Displacement of αCT from the L1 domain induces m-pY1150
In the apo-IR structure, L1 forms a complex with the αCT′ helix of the other protomer, which acts as the primary binding site for insulin binding[19,20,32,33]. A62 binding on the L1 surface competitively blocks αCT′ binding to L1, resulting in the displacement of αCT′ from L1, which is disordered in the structure (Fig. 1f). S519, an insulin mimetic peptide, is a selective agonist for IR, like A62, and the C-terminal 16 residues of S519 (S519-C16) also competitively binds to the same site (αCT′) on the L1 surface[14,34,35]. Moreover, we verified that S519-C16 also stimulates only m-pY1150 (Fig. 1g). To demonstrate the importance of displacement of αCT′ from L1 for m-pY1150, we replaced Phe705 of αCT′ to Ala (F705A) to disrupt the interaction between αCT′ and L1[32]. The F705A mutation significantly enhanced the potency of A62. By contrast, Tyr and Trp substitutions of Arg702 and Thr704 (R702Y/T704W) in αCT′ that augmented the interaction between L1 and αCT′ significantly inhibited A62 activity (Fig. 1h)[34]. These results suggest that the arrowhead conformation of IR induced by the displacement of αCT′ from the L1 domain is a key step in m-pY1150 formation and for selective activation of the metabolic response.

### One A62 can bind to IR asymmetrically with one insulin
The A62 aptamer not only acts as an agonist, but also positively regulates the binding of insulin to IR at low concentrations[18]. The structure of the IR$_{A62+Ins}$ complex is asymmetric, with one insulin and one A62 bound at each site of the dimer (Fig. 2a). While the A62-bound head retains the overall conformation observed in IR$_{2xA62}$, insulin binding lifts half of the arrowhead comprised of the L1′-CR′-L2′ module, and folds L1′-CR′ + αCT toward FnIII-1, resulting in a T with a tilted head (referred to as tilted T-shaped) conformation (Fig. 2a). Insulin binds at the site comprised of L1′, αCT, and FnIII-1, which is virtually identical to the single insulin-bound structure[22,23]. Although single insulin binding rearranges the conformation of the insulin-binding module, A62 binding at the opposite site blocks the additional conformational changes that are required for full phosphorylation of IR.

The A62 aptamer consists of 25 nucleotides with various modified bases (Supplementary Fig. 3a). A62 forms a non-helical compact structure, which is primarily stabilized through numerous base

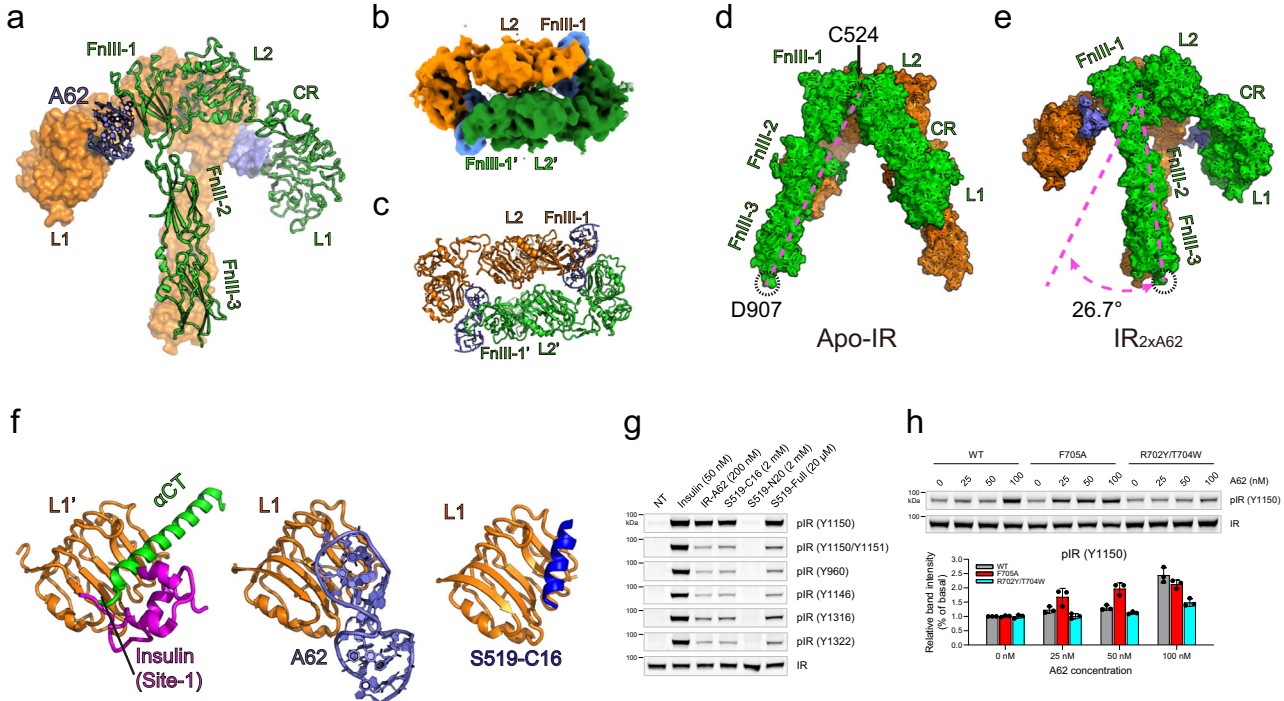

**Fig. 1 | Cryo-EM structures of A62-bound IR intermediates. a** Structure of $IR_{2xA62}$, with one protomer (orange) shown in surface representation and another protomer (green) in ribbon representation. A62 aptamers are colored violet. **b** Surface representation of the $IR_{2xA62}$ complex in the top view. Each protomer is shown in orange or green. The A62 aptamers are colored violet. **c** Ribbon representation of Fig. 1b in the same orientation. **d, e** Comparison of the structures between apo-IR (**d**; PDB: 4ZXB)[33] and $IR_{2xA62}$ dimers (**e**). The angle of rigid body rotation was measured as the angle change of Asp907 centered on Cys524 of each protomer. **f** Structure of A62 binding competitively to L1 with insulin and the αCT domain. The close-up view shows the αCT + insulin+L1 complex (left) and the A62 + L1 complex (middle) of the $IR_{2xA62}$ structure. The S519-C16 peptide (blue; PBD: 5J3H, right)[35]

binds to L1 (orange) similarly to αCT. **g** Displacement of the αCT domain from the L1 domain preferentially induces m-pY1150 formation. Site-specific IR phosphorylation was performed in Rat-1 cell lines stably expressing human IR. Cells were stimulated with the indicated peptide or A62 for 1 h or insulin for 5 min. The data were representative of three independent experiments. **h** Enhancement of A62 aptamer activity by displacement of αCT from L1. A62-induced IR phosphorylation (m-pY1150) was measured in CHO-K1 cells expressing wild-type (WT) IR and its F705A and R702Y/T704W mutants. Cells were stimulated with A62 for 1 h. Each experiment was repeated three times independently, and the graph is presented as mean ± standard deviation of replicates ($n = 3$). Source data are provided as a Source Data file.

stacking interactions and five Watson-Crick (WC) base pairs along with hydrogen bonds and hydrophobic interactions (Fig. 2b, c). Modified bases (2Nap10, 2Nap20, and 5Bz14) occupy the space between the multilayer stacking interactions and contribute to an unwinding of an aptamer, resulting in the formation of three loops; a head loop (loop H, fA12 to dA17), a short (loop S, dG18 to fC22), and a long loop (loop L, dG7 to mG11) at each side, as well as a stem (fA2 to fC6). The A62 structure is described in detail in the Supplementary note and Supplementary Fig. 3.

## A62 binding mimics site-1/site-2′ insulin coordination

The elongated A62 aptamer fills the space between L1 and the side of FnIII-1′ domains vertically relative to the stalk, and bridges the two domains through its loops (Fig. 2d, e). In the L1 site, loops H and L of A62 make contact with L1-β2. This interface is stabilized through extensive hydrophobic interactions, ion pairs, and H bonds between A62 bases and IR residues (Fig. 2b, f). On the opposite side, a flat face formed by the stem and two loops is packed against the side of the main β-sheet of FnIII-1′ through stacking between modified bases and β-sheet residues, and ion pairs between the A62 phosphate backbone and basic residues (Fig. 2b, g).

To confirm the functional significance of the A62 bridge between L1 and FnIII-1′, we mutated residues at the A62-binding interface at L1 (Arg14, Lys40, Phe64, and Arg65) and FnIII-1′ (Tyr477, Arg479, Arg488, and Arg554). All mutations dramatically reduced m-pY1150 of IR induced by A62, suggesting that the A62-mediated crosslink between L1 and FnIII-1′ is critical for Y1150 phosphorylation of IR (Fig. 2h, I).

The binding sites of A62 on L1 and FnIII-1′ overlap with the previously identified insulin-binding site-1 and site-2′, respectively (Fig. 2d)[22,23,27]. Moreover, the A62 bridge is almost identical to that reported for site-1/site-2′ insulin coordination (Supplementary Fig. 4c–e)[36]. However, unlike A62, that interacts with site-1 and site-2′ simultaneously, structures of IR complexed with two or three insulin molecules revealed that insulin in site-1/site-2′ is coordinated in two different orientations (site-1- or site-2′-biased orientation), and cannot stably bind to both site-1 and site-2′ simultaneously[36]. Consequently, the distance between L1 and FnIII-1′ in $IR_{A62+Ins}$ is closer than the site-1/site-2′ insulin coordination in the reported structures (Supplementary Fig. 4f–h). Therefore, the A62 bridge seems to trap the intrinsically unstable IR intermediates by mimicking the site-1/site-2′ insulin coordination.

## A43 binding to IR stabilizes the single insulin-bound Γ-shaped structure

In contrast to A62, the A43 aptamer alone has no agonistic activity, but it acts as a positive allosteric modulator for IR activation in the presence of insulin[30]. Moreover, A43 and insulin binding to IR exhibited mutual positive cooperativity[30]. To understand how A43 stabilizes the insulin-bound IR, we prepared the complex of the IR dimer, A43, and insulin ($IR_{A43+Ins}$), and determined the structure at 3.62 Å resolution (Supplementary Fig. 5a–g). One A43 aptamer and one insulin bind at each site of the IR dimer, forming an asymmetric Γ-shaped structure (Fig. 3a–c). The central feature of the $IR_{A43+Ins}$ complex is the presence of αCT helices bound to the L1 domain in both A43- and insulin-bound

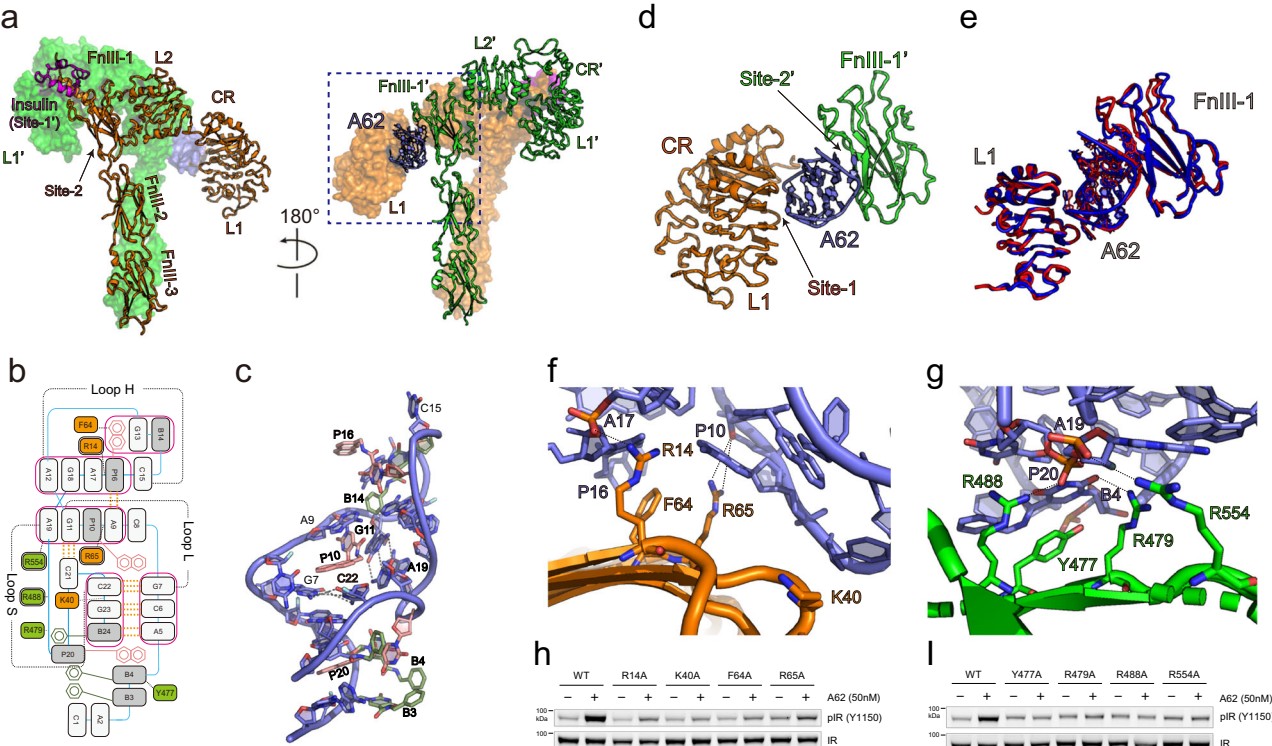

**Fig. 2 | Cryo-EM structure of the IR$_{Ins+A62}$ complex. a** Half-turn views of the IR$_{A62+Ins}$ structure. Each promotor (orange or green) of the dimer is shown in surface representation or ribbon representation. The A62 aptamer is colored violet and the insulin is colored magenta. **b** Cartoon representation of the A62 structure. Labels for the modified parts on the 2′ carbon in the ribose sugar (d, f, m) are omitted for clarity. PX and BX represent (5-[*N*-(1-naphthylmethyl)carboxamide]-2′-deoxyuridine) and (5-[*N*-benzylcarboxamide]-2′-deoxyuridine), respectively. X is the nucleotide number. The A62-interacting residues are shown; the stacking interactions are shown in pink boxes, base pairings are shown in yellow dots, the FnIII-1′-base interactions are in green boxes, the CR and L2 residue-base

interactions are in orange boxes, and the residue-phosphate interactions are in double-lined boxes. **c** Overall structure of the A62 aptamer. **d** Close-up view of the A62 bridge across L1 and FnIII-1′ (the blue box in Fig. 2a). **e** Superimposed structure of the A62 bridge across L1 and FnIII-1′ from IR$_{A62+Ins}$ (blue) and IR$_{2xA62}$ (red). **f, g** Close-up view of the binding interface between A62 and IR on **f** L1 and **g** FnIII-1′ domains. **h, I** A62-induced IR phosphorylation (m-pY1150) in CHO-K1 cells expressing WT IR or the indicated point mutants predicted to disturb A62 binding to **f** L1 or **g** FnIII-1′ domains. Cells were stimulated with 50 nM A62 for 1 h. The data were representative of three independent experiments. Source data are provided as a Source Data file.

sites. The two αCT helices form a linearly elongated bridge in the opposite direction to their C-terminal tails, connected by a disulfide bond on the insert domains (Fig. 3b). The similar αCT-αCT′ bridge was observed in the structure of IGF1R complexed with IGF-1[24].

The A43 aptamer consists of 31 nucleotides in which six dTs are substituted by Ps (Supplementary Fig. 6a). A43 folds into a highly compact, helical turn structure with dimensions of 48 × 21 × 20 Å. The aptamer can be divided into a 13 nucleotide stem (dT4 to dP8 and dP24 to dC31), a four nucleotide turn (dA22-dP23 and dC9-dC10), and an 11 nucleotide loop (dG11 to dC21). The structure of the A43 aptamer is maintained primarily through numerous multilayer base-base stacking interactions (Fig. 3d, e). The A43 structure is described in detail in the Supplementary note and Supplementary Fig. 6a–d.

A43 fits snugly into the site formed by CR, L2, and FnIII-1′ opposite the insulin-binding module (Fig. 3c). We introduced Ala substitutions to residues at the A43-binding interface of CR (Arg271), L2 (Ser323, Thr325), and FnIII-1′ (Tyr477, Lys484, Leu486, Arg488, Trp551, Leu552, and Arg554). All mutations dramatically reduced the insulin-enhancing activity of A43, which indicates that the fitting of A43 into multiple domains is critical to stabilize insulin binding to IR (Supplementary Fig. 6e–k). Moreover, the IR$_{A43+Ins}$ structure is nearly identical to the single insulin-bound IR structure (Supplementary Fig. 7a, b)[22,23]. Thus, these results indicate that A43 potentiates the single insulin binding of the Γ-shaped conformation by stabilizing the interaction between αCT′ and L1 opposite the insulin-binding module, and single insulin binding to IR is sufficient to fully activate it.

During the classification of the IR$_{A43+Ins}$ dataset, we observed a population of particles (23% of total particles) with a symmetric T-shaped conformation (4.18 Å, Supplementary Figs. 5a, 8a–e). Only two insulin molecules but no aptamer associated with the T-shaped IR (Supplementary Fig. 8e, f). Each insulin bound to site-1 and site-1′, which suggests that the second insulin competes with A43 in binding to IR (Supplementary Fig. 8e). Overall structure of the T-shaped IR is similar to the reported structure of the two- or four-insulin-bound IR complex with root mean square deviation in the range of 2.18 Å (6PXV and 6SOF)[21,27] to 2.3 Å (7STH)[36] for entire Cα atoms in the ectodomain (Supplementary Fig. 8f–h).

### Blocking the second insulin binding to IR can potentiate IR activation

Structural comparison of apo-IR and IR$_{A43+Ins}$ by aligning the CR domains showed that binding of A43 to the apo-IR structure involved significant steric crashes with L2 and FnIII-1′ (Supplementary Fig. 7c)[20,33]. This indicates that the insulin-bound Γ-shaped structure of IR is more favorable for A43 binding than the apo-IR structure, which explains the mutual positive cooperativity between insulin and A43[23,24,30]. In contrast to insulin, which exhibits negative cooperativity, A62 and A43 exhibit positive cooperativity for insulin binding to IR[18,30]. A62 and A43 share a common feature that binds to the same site with site-2 on FnIII-1 (Fig. 3f). We predict that the aptamers block the binding of second insulin to the opposite site, which prevents negative cooperativity by the second insulin, and the aptamers thereby protect binding of the first insulin (Fig. 3g).

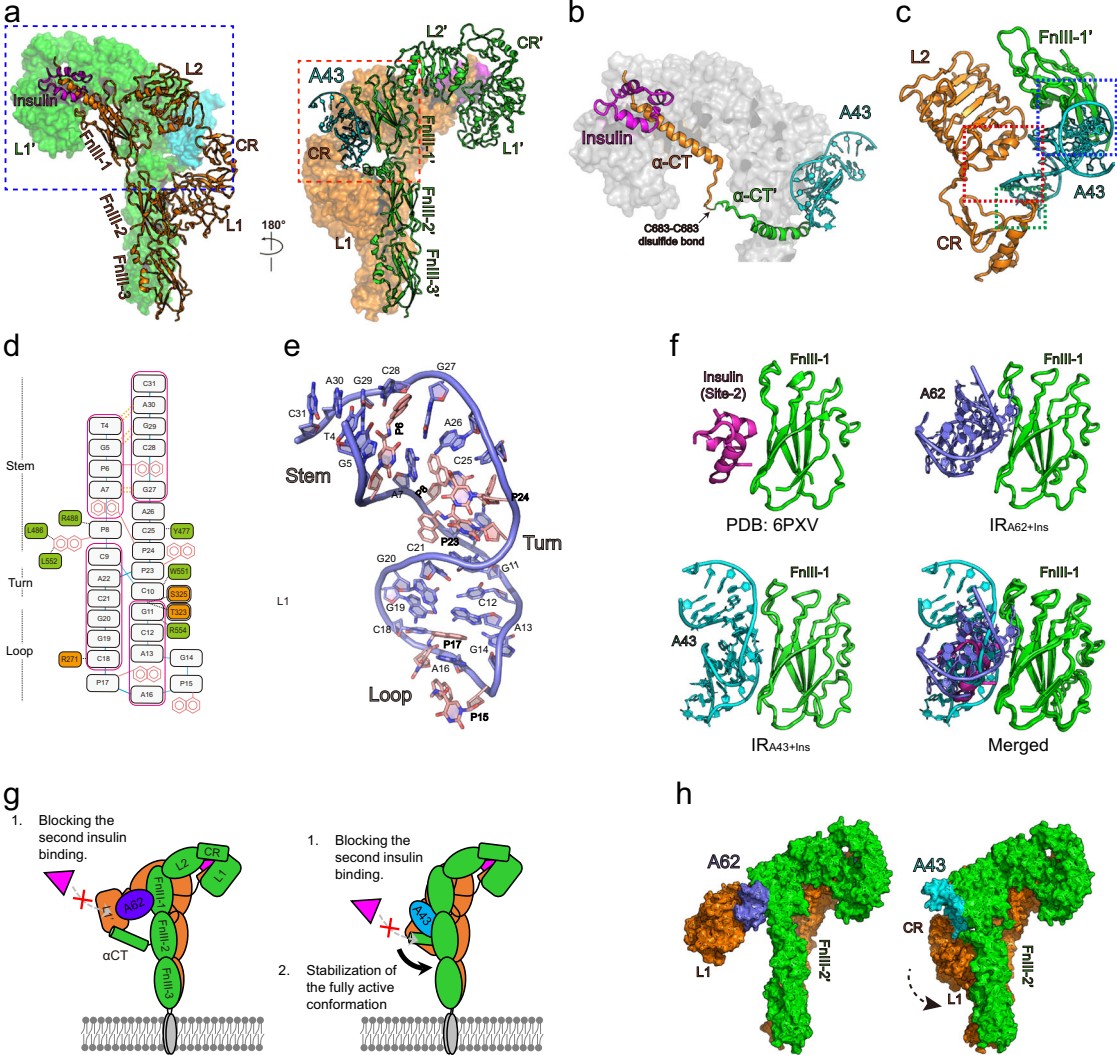

**Fig. 3 | Cryo-EM structure of the IR_Ins+A43 complex. a** Half-turn views of the IR_A43+Ins structure. Each protomer of a dimer is colored green or orange. The A43 aptamer is colored cyan and the insulin is colored magenta. **b** Close-up view of the αCT-αCT′ crosslinked bridge between insert domains (blue box in Fig. 3a). **c** Close-up view of the A43 binding site located at the CR, L2, and FnIII-1′ domains (red box in Fig. 3a). **d** Cartoon representation of the A43 structure. Labels for the 2′-deoxy-ribose sugar (d) are omitted for clarity. The A43-interacting residues are shown in the same scheme as in Fig. 2b. **e** Overall structure of the A43 aptamer. **f** Structure of insulin binding to FnIII-1′ (top left, PBD: 6PXV)[27]. Structure of A62 binding to FnIII-1′ from the IR_2xA62 complex (top right). Structure of A43 binding to FnIII-1′ from the IR_A43+Ins complex (bottom left). **g** Cartoon representation of a model for the positive cooperativity of aptamers with insulin binding. **h** Structural comparison of IR_A62+Ins (left) and IR_A43+Ins (right) shown in surface representation. A62 and A43 are colored violet and cyan, respectively.

The most significant structural difference between IR_A43+Ins and IR_A62+Ins is the position of the L1 domain opposite to the insulin-binding site (Fig. 3h). Structural comparison of IR_A62+Ins and IR_A43+Ins by aligning the insulin-bound module showed that the L1 and CR domains in the A43-binding site undergo rigid body rotation by 70° toward the FnIII-1 domain in the A62-binding site. Release of A62, which imposes a strain on L1, allows the translocation of L1 and CR domains, along with αCT′, toward the FnIII-2 stalks.

**The distance between the FnIII-3 ends determines the transition from m-pY1150 to full phosphorylation**

In IR_2xA62, IR_A62+Ins and IR_A43+Ins structures, ligand binding brings two FnIII-stalks almost parallel. However, the distances between the FnIII-3 ends (Asp907) in the IR_2xA62 and IR_A62+Ins structures are significantly longer than in IR_A43+Ins (Fig. 4a). IR_A43+Ins has one major structure in which the FnIII-3 ends are 26 Å apart. By contrast, the structural states of FnIII-stalks in both IR_2xA62 and IR_A62+Ins are heterogeneous, and classified into three and two different states, respectively, in which the distances between FnIII-3 ends vary from 34 to 45 Å (Supplementary

Fig. 9a–c). The similar distance range between the FnIII-3 ends of IR_2xA62 and IR_A62+Ins suggests that A62 binding prevents the FnIII-3 ends from being as close as 26 Å, even in the presence of insulin.

In IR_A43+Ins, L1 interacts with FnIII-2 and FnIII-2′, and complex formation brings FnIII-3 and FnIII-3′ into close proximity (Fig. 4b–d). Moreover, A43 binding to IR potentiates the phosphorylation of all Tyr residues of IR induced by insulin[30]. Therefore, we surmised that the small difference in the distance between parallel FnIII-stalks regulated by the L1 and FnIII-2/FnIII-2′ interaction is a critical structural factor determining the transition from m-pY1150 to full activation (Fig. 4d). To investigate the role of the translocation of L1 opposite the insulin binding module, we introduced F64A or R702Y/T704W mutations, which we expected to disrupt or augment the L1−αCT′ interaction, respectively, into a single protomer[22,23,34,37]. We also attached yellow fluorescent protein (YFP) to the C-terminus of one protomer, and transfected a 1:1 mixture of YPF-tagged IR and untagged IR to distinguish the phosphorylation state of each protomer (Fig. 4e and Supplementary Fig. 10a). While insulin fully activated the wild-type (WT) + WT/YFP hybrid receptor, both F64A and R702Y/T704W

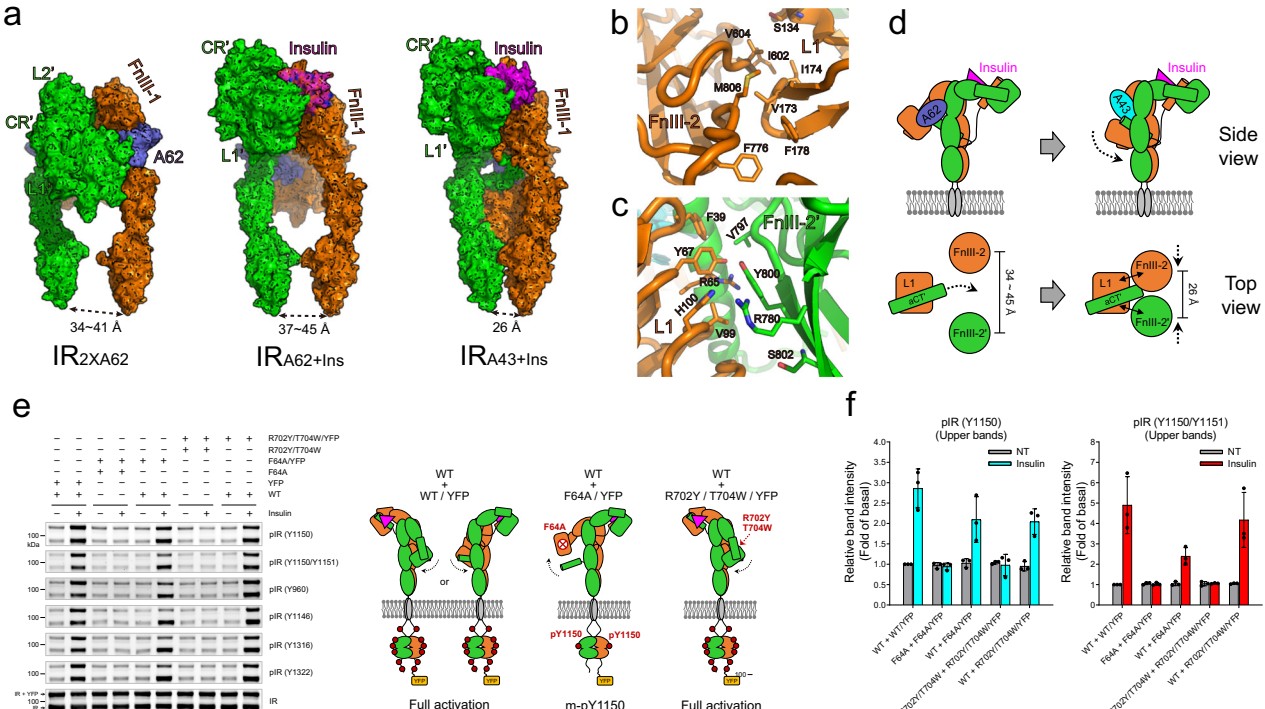

**Fig. 4 | The position of the non-insulin-bound L1 domain determines the autophosphorylation state of IR. a** Comparison of the $IR_{2xA62}$ (left), $IR_{A62+Ins}$ (middle) and $IR_{A43+Ins}$ (right) structures. Side views of the structures are shown in surface representation to highlight the arrangement of FnIII-3 stalks. Each protomer is colored green or orange. A62 is colored violet and insulin is colored magenta. The distance between the membrane-proximal ends (Asp907) of the stalks is highlighted. **b, c** Interaction of the L1 domain (orange) with **b** FnIII-2 (orange) and **c** FnIII-2′ (green) in the $IR_{Ins+A43}$ structure. **d** Cartoon representation of a model for distance regulation between parallel FnIII-stalks by translocation of the L1 domain. **e** Insulin-induced IR phosphorylation in CHO-K1 cells expressing hybrid IR consisting of short-IR (untagged, lower bands) and long-IR (YFP-tagged, upper bands). Cells were stimulated with 200 nM insulin for 5 min. (YFP: yellow fluorescent protein.) The data were representative of three independent experiments. **f** Quantification of western blot data for m-pY1150 or pY1150/pY1151 shown in Fig. 4e. Bar graphs are presented as means ± standard deviation of replicates ($n = 3$). Source data are provided as a Source Data file.

mutations completely inhibited IR phosphorylation by disrupting insulin binding (Fig. 4e, f). In the WT + F64A/YFP hybrid receptor, asymmetric insulin binding induced m-pY1150 in YFP-tagged IR (upper bands) to the same degree as in the WT + R702Y/T704W/YFP hybrid receptor. However, phosphorylation of other Tyr residues was significantly reduced only in the WT + F64A-YFP hybrid receptor in which the L1–αCT′ interaction opposite the insulin-binding module is disrupted.

Moreover, to disrupt the L1–FnIII-2/FnIII-2′ interaction, we replaced Val99, Val173, Val604, and Ser802 at L1 and FnIII-2 with Arg (V99R/V173R/V604R/S802R). These mutations did not affect m-pY1150 significantly, but they reduced the phosphorylation of other Tyr residues (Supplementary Fig. 10b–d). These results suggest that translocation of the L1 head toward FnIII-2/FnIII-2′ is required for the structural transition of the m-pY1150 state (FnIII-3 ends distance 34–45 Å) to fully phosphorylated IR (FnIII-3 ends distance 26 Å).

### Interaction between JM and kinase domains prevents full phosphorylation of IR

One plausible explanation for the above results is that restriction of the distance between TM or kinase domains controlled by FnIII-3 ends prevents the transition from m-pY1150 to full activation. To test this hypothesis, we introduced a flexible linker ($IR_{Linker}$, GGGGSGGGGS) between FnIII-3 (Leu909) and TM (Asp910) domains to release the constraint between the two elements. In the $IR_{Linker}$ mutants, m-pY1150 stimulated by A62 was increased compared with WT, but the transition to full activation was not observed (Fig. 5a, b). Furthermore, insulin stimulation of $IR_{Linker}$ mutants induced m-pY1150 to the same degree as observed for WT, whereas phosphorylation of other Tyr residues was significantly reduced (Supplementary Fig. 10e–g). The previous

study suggests that the TM dimerization induced by the ligand binding is important for the autophosphorylation of kinase domains[25]. The flexibility between FnIII-3 and TM helix may disallow TM helices to dimerize and prevents the transduction of conformational changes in the extracellular domain to the intracellular domain. These results suggest that close (~26 Å) apposition of the FnIII domains followed by additional conformational rearrangement of intracellular domains are required for full (multi) phosphorylation of the cytoplasmic domains, but that mono-phosphorylation of Y1150 can occur at intermediate FnIII distances (34–45 Å).

The intracellular JM domain of IR represses its kinase activity by interacting with the N-terminal lobe of the kinase domain through Tyr972, and it plays a role in the conformational rearrangement of IR kinases[38–41]. To understand whether Tyr972 is involved in the transition from m-pY1150 to full activation, we disrupted the interaction between JM and the kinase domains by replacing Tyr972 with Ala. The Y972A mutation did not dramatically change m-pY1150 stimulated by insulin, but it increased the phosphorylation of other Tyr residues nearly twofold compared with WT. Moreover, in the Y972A mutants, the selective stimulation of m-pY1150 by A62 disappeared, and we instead observed phosphorylation of all Tyr residues, as observed for insulin binding (Fig. 5c, d). This suggests that the interaction between JM and kinase domains restricts the transition from m-pY1150 to full activation.

### Discussion
In this work, we determined three structures of aptamer-bound IR complexes with kinase domains in different phosphorylation states: mono-phosphorylated arrowhead-shaped $IR_{2xA62}$ and tilted T-shaped $IR_{A62+Ins}$, and the fully phosphorylated Γ-shaped $IR_{A43+Ins}$. Both

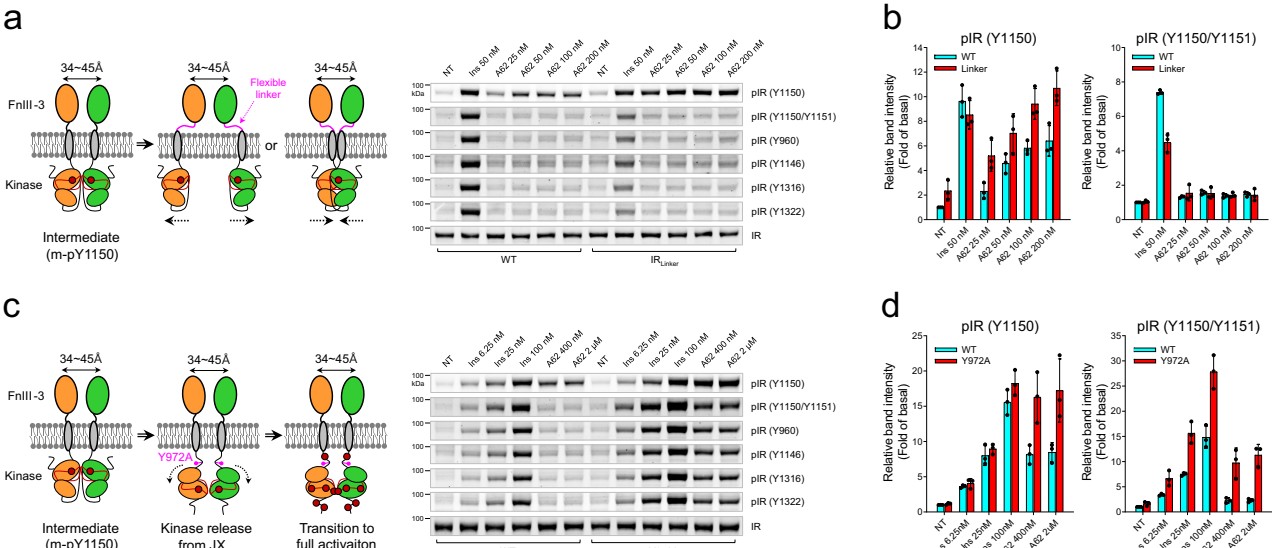

**Fig. 5 | The JM domain plays a key role in preventing full activation. a** Insulin- or A62-induced IR phosphorylation in CHO-K1 cells expressing WT IR or mutant IR with a flexible linker (IR$_{Linker}$). **b** Quantification of western blot data for m-pY1150 or pY1150/pY1151 shown in Fig. 5a. **c** Insulin- or A62-induced IR phosphorylation in CHO-K1 cells expressing WT IR or its Y972A mutant. **d** Quantification of western blot data for m-pY1150 or pY1150/pY1151 shown in Fig. 5c. **a, c** The data were representative of three independent experiments. **b, d** bar graphs are presented as means ± standard deviation of replicates (n = 3). Source data are provided as a Source Data file.

arrowhead- and tilted T-shaped IR conformations represent a state(s) for the selective activation of the metabolic pathway. In these conformations, the distance between the membrane-proximal ends of FnIII-3 is in the range of 35 to 45 Å and only Y1150 became phosphorylated. A62 simultaneously binds to the site-1(L1) and site-2'(FnIII-1') of IR, and traps an unstable conformation in which αCT' is displaced from L1, preventing the translocation of L1 to FnIII-2 and FnIII-2'. These structural features in the extracellular domain may be responsible for the selective mono-phosphorylation on Y1150.

Induction of the m-pY1150 state of IR by A62 suggests that the distance of 34–45 Å between the two FnIII-3 ends is close enough to trigger *trans*-mono-phosphorylation of two IR kinases, but not full phosphorylation. Theoretically, phosphorylation of Y1146 and Y1151 located close to Y1150 may also occur in *trans* at the same distance, but our results showed that this does not happen. A plausible explanation is that Y1150 is the most efficient substrate among the three tyrosine residues in the activation loop, and the specificity of Y1146 and Y1151 may be lower than Y1150 for *trans*-autophosphorylation of IR kinase. The crystal structure of the IR kinase-peptide complex (PDB: 1IR3) showed that the residue preceding (P-1, where P is the acceptor tyrosine pY) Tyr engages in a water-mediated hydrogen bond with a positively charged residue[42]. The two hydrophobic residues following pY (Met at P + 1 and P + 3) are located in hydrophobic pockets. The P-1, P + 1, and P + 3 residues for Y1150 are Asp, Tyr, and Lys, respectively. The three equivalent residues for Y1146 are Ile, Glu, and Asp, and those for Y1151 are Tyr, Arg, and Gly. This suggests that Y1146 and Y1151 are less favorable compared with Y1150 in terms of substrate preference for IR kinase. We suggest that the Y972 *cis*-autoinhibitory interaction with the kinase domain prevents facile *trans*-phosphorylation of all but Y1150 (which still requires insulin) and that maintaining the kinase domains in close proximity via the L1–FnIII-2/-2' interaction (~26 Å) is necessary to achieve full phosphorylation.

Because Y1150 is the earliest phosphorylated Tyr in the kinase domain, it is reasonable to speculate that the arrowhead- and tilted T-conformations represent intermediate states[43–45]. In the inverted V-shaped apo-IR conformation, insulin cannot access the primary binding sites consisting of L1 and αCT' interfaces due to steric crashes

with the nearby FnIII-2' domain[33]. Thus, initial insulin binding to IR requires a structural transition from the inverted V conformation. Our study suggests that the arrowhead-shaped IR$_{2xA62}$ is a plausible candidate in which insulin binding can occur by simultaneously engaging with site-1 and site-2'. Simple rigid body rotation of each protomer allows the inverted V to adopt the arrowhead conformation (Fig. 1d, e and Supplementary Fig. 1m). In addition, lifting an L1 arm rearranges the arrowhead-shaped IR into the tilted T conformation. Importantly, both arrowhead- and tilted T-shaped IR share a similar distance between FnIII-3 ends and the mono-phosphorylation state. Based on these features, we propose that the arrowhead conformation represents a state between inverted V and tilted T-shaped IR conformations. Because the extracellular domain of IR displays a high degree of structural heterogeneity in the absence of insulin[22,23], the inverted V-shaped and arrowhead conformations of IR may be present in equilibrium in the absence of insulin, which allows insulin to access site-1 and site-2' simultaneously by exposing the L1 and αCT' interfaces (Fig. 6a, b). A crystal structure of partial IR domains (L1, CR, L2, FnIII-1, and αCT) also displayed an insulin-bound symmetric arrowhead conformation similar to the head domain of IR$_{2xA62}$, supporting our model[32].

The initial binding of insulin to site-2' has been proposed previously[29]. We do not exclude such a possibility. In that case, insulin may bind to site-2' of the inverted V-shaped apo-IR, and subsequently interacts with both site-1 and site-2' of the arrowhead-shaped IR. However, mutation of residues that disturb site-2 only partially reduced IR autophosphorylation, and the binding affinity of insulin to site-2' was too low (Kd ~400 nM) relative to physiological insulin concentrations[27,46]. Moreover, we demonstrated that asymmetric insulin binding to one site-1 of IR can stimulate IR autophosphorylation in multiple Tyr residues (Fig. 4e). Thus, single insulin-bound Γ-shaped IR conformations may represent the fully-active state of IR. In the Γ-shaped conformation, one insulin interacts with only site-1 of IR, while site-2 is not involved in insulin binding. Therefore, we suggest that site-2' plays an auxiliary role in the initial binding of insulin. After the initial insulin binding occurs under site-1/site-2' coordination, insulin may subsequently dissociate from site-2', and the insulin and site-1 module are lifted to form the fully-active Γ-conformation.

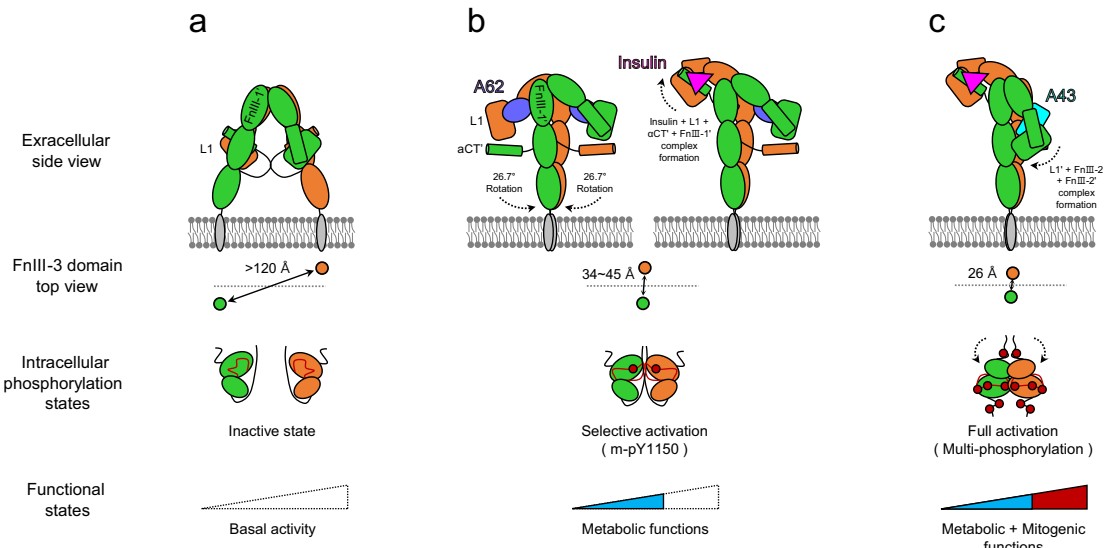

**Fig. 6 | Cartoon representation of a proposed model for selective activation of IR. a** Apo state with basal activity; **b** Two m-pY1150 states, arrowhead-shaped IR$_{2xA62}$ (left) and the tilted T-shaped IR$_{A62+Ins}$ (right), that selectively activate the metabolic signaling. A62 and insulin are shown in a purple circle and pink triangle, respectively. The primary insulin binding site in the arrowhead conformation can be exposed by the rigid body rotation of the apo-IR in a. The site-1+insulin module is lifted up toward FnIII-1′ (right). Initial insulin binds to site-1 and site-2′; **c** In the IR$_{A43+Ins}$ structure, L1′ translocation toward FnIII-2 stalks for full activation. A43 is shown in the cyan square.

In summary, we propose a hypothetical model in which the initial insulin binding to IR transiently induces the arrowhead and the tilted T IR conformations. Given the functional selectivity of A62 and S519 insulin mimetic peptides, the intermediate states of IR may be responsible for selective stimulation of the AKT pathway and the metabolic functions of IR (Fig. 6b). In the fully-active state, translocation of the L1 head toward FnIII-2/FnIII-2′ brings the two FnIII-3 ends close to 26 Å, which promotes full phosphorylation and the mitogenic functions of IR (Fig. 6c). Our findings suggest that conformational change of FnIII-3 (or extracellular domain) must be precisely relayed to TM domains and intracellular domains for autophosphorylation of the kinase domains. We showed that some conformational rearrangement of intracellular domains, such as the release of JM domains from kinases, is also required for the transition to full phosphorylation.

Recently, IR structures with a tilted T-shape similar to that of IR$_{A62+Ins}$ have been reported[28,29,36]. Unlike the IR$_{A62+Ins}$ structure, these structures exhibit two or three insulin-bound IRs. Although IR binds two to four insulins at extremely high insulin concentrations >100 nM, IR binds only one insulin with high affinity at physiological insulin concentrations (pM). However, the binding of the second insulin to IR accelerates the dissociation rate of pre-bound insulin (negative cooperativity)[39]. Moreover, the IR$_{A43+Ins}$ structure verified that the position of the L1 domain opposite the bound insulin is critical for the fully-active state of IR and the enhancement of insulin binding. Thus, it is likely that the tilted T-shaped IR conformation represents a late IR intermediate in which the second insulin causes negative cooperativity. However, dose-response curves for the dissociation of pre-bound insulin are bell-shaped and reveal a loss of dissociation acceleration at high insulin concentrations >100 nM[26]. We suggest that the binding of third and fourth insulin to site-2 and site-2′ of IR protects the pre-bound insulin by preventing the translocation of the pre-bound insulin module.

Functionally selective peptide-based or antibody agonists for IR have been proposed as potential anti-diabetic agents with a low risk of vascular complications and for tissue-selective actions[15,16]. Our current work provides a framework for designing alternative agonists comprised of nucleotides for IR, selectively potentiating insulin activity for diabetes treatment. Our work also suggests that aptamers can be applied to stabilize otherwise unstable conformations of proteins, making them accessible for structural studies to reveal atomic details.

## Methods

### Cloning and establishing stable cell lines

The short isoform (Isoform A) of human IR followed at its C-terminus by a 16-residue TGHHHHHHDYKDDDDK sequence (i.e., *Age*I restriction site, 6xHis tag, and FLAG tag) was cloned into the pcDNA3.1 mammalian expression vector (Invitrogen, Carlsbad, CA, USA). FreeStyle HEK293F cells were maintained in Freestyle293 medium (Thermo Fisher Scientific, Waltham, MA, USA) at 37 °C with shaking at 90 rpm and 8% $CO_2$. Before transfection, the medium was replaced with fresh medium, and 30 ml of cells ($1 \times 10^6$ cells/ml) were transfected with 37.5 μg of IR-containing pcDNA3.1 vector using polyethylenimine (PolyScience, Niles, IL, USA) at a ratio of 3:1 (w/w) with DNA. At 24 h post-transfection, the selection was undertaken in the presence of 500 μg/ml Geneticin (Gibco) for the generation of a stable cell line expressing the protein. To remove dead cells during the selection process, the medium was replaced with fresh medium containing 500 μg/ml Geneticin every 24 h for 3 weeks. After cell viability was completely restored to the level before selection, the expression level of IR in the stable cell line was analyzed by sodium dodecyl sulfate-polyAcrylamide gel electrophoresis (SDS-PAGE) and western blotting. Detailed information on reagents is provided in Supplementary Table 2.

### Expression and purification of IR$_{2×62}$, IR$_{A62+Ins}$, and IR$_{A43+Ins}$ complexes

For expression of apo-IR, stable cells were grown in suspension in Freestyle293 medium at 37 °C and 8% $CO_2$. At a cell density of ~4.0 × 10⁶ cells/ml, cells were collected by centrifugation and resuspended in a buffer comprising 20 mM HEPES pH 7.5, 400 mM NaCl, 200 nM recombinant human insulin (RHI), and 5% glycerol. Cells were incubated on ice for 1 h, disrupted using a Dounce Homogenizer (Kimble) on ice, and solubilized in buffer comprising 20 mM HEPES pH 7.5, 400 mM NaCl, 1 mM EDTA, 1% (w/v) *n*-dodecyl β-ᴅ-maltoside (DDM; Anatrace), 0.1% (w/v) cholesterol hemisuccinate (CHS; Sigma), 5% glycerol, 1 mM EDTA, 200 nM RHI, and protease inhibitor cocktail (Roche) for 2 h. For purification of IR complexed with A62 alone,

500 nM A62 aptamer, 5 mM KCl, and 5 mM MgCl$_2$ replaced RHI in the solubilization buffer. Solubilized membranes were isolated by ultra-centrifugation with a Ti45 rotor (Beckman) at 100,000×g for 1 h at 4 °C. The supernatant was isolated and applied to anti-Flag affinity G1 resin (GenScript) for 2 h at 4 °C. The resin was washed in a batch with washing buffer (20 mM HEPES pH 7.5, 400 mM NaCl, 5% glycerol, 0.1% DDM, 0.01% CHS, and 200 nM RHI) for IR$_{A62+Ins}$ and IR$_{A43+Ins}$ purification. Proteins were eluted using elution buffer (20 mM HEPES pH 7.5, 400 mM NaCl, 5% glycerol, 0.03% DDM, 0.003% CHS, 200 nM RHI, and 0.4 mg/ml Flag peptide). To remove flag peptide and insulin, eluted proteins were concentrated using an Amicon Ultra centrifugal device (100 kDa cut-off; Millipore) and diluted with buffer comprising 20 mM HEPES pH 7.5, 400 mM NaCl, 5% glycerol, 0.03% DDM, 0.003% CHS, 5 mM KCl, and 5 mM MgCl$_2$. For IR$_{2xA62}$, 200 nM RHI was replaced by 500 nM A62 in the washing and elution buffer. Proteins were concentrated using an Amicon Ultra centrifugal device (100 kDa cut-off; Millipore). To generate the complex with aptamer, proteins and pre-activated aptamers (A43 or A62) were mixed at a molar ratio of 1:2 and incubated for 1 h on ice. The mixture was injected onto a Superose 6 10/300 column equilibrated with buffer comprising 20 mM HEPES pH 7.5, 105 mM NaCl, 5 mM KCl, 5 mM MgCl$_2$, 0.03% DDM, and 0.003% CHS. Eluted fractions were pooled and concentrated to 8 mg/ml using a Vivaspin device (100 kDa cut-off; GE Healthcare) for cryo-EM analysis.

## Cryo-EM sample preparation and data collection

To prepare cryo-EM grids, 3 μl of the sample was applied to glow-discharged holey carbon grids (C-flat 1.2/1.3 Au 400-mesh; EMS). Grids were plunge-frozen in liquid ethane using a Vitrobot Mark IV (Thermo Fisher Scientific) with a blot force of 4 for 5 s at 100% humidity and 4 °C. For IR$_{2xA62}$ and IR$_{A43+Ins}$ complexes, images were acquired using a Talos Arctica electron microscope (FEI) operated at 200 kV and equipped with a Gatan K3 Summit direct electron detector in counting mode (at the Photon Science Center, Pohang University of Science and Technology) at a nominal magnification of 79,000×. Movies were collected comprising 10,960 micrographs for IR$_{A43+Ins}$ and 8856 micrographs for IR$_{2xA62}$. Datasets were collected with a pixel size of 1.07 Å and a defocus of −1.5 to −3.0 μm.

For the IR$_{A62+Ins}$ complex, two datasets were collected using a Titan Krios G4 instrument operated at 300 kV and equipped with a Gatan K3 Summit direct electron detector in fast mode at a nominal magnification of 105,000×. For the first dataset (dataset 1), 11,400 movies were collected. After several rounds of particle sorting, we found out that the selected number of particles were not sufficient to produce high-quality map. Therefore, we collected additional 12,930 movies (dataset 2) from another grid. All movies were collected with a pixel size of 0.85 Å and a defocus of −0.5 to −2.25 μm. Micrographs were dose-fractionated over 50 frames with an accumulated dose of 50 electrons per Å$^2$.

## Data processing for the IR$_{2xA62}$ complex

For the IR$_{2xA62}$ complex, dose-fractionated image stacks from 8,856 movies were imported to CryoSPARC v3.3.1[47]. The imported images were subjected to dose-weighting using full-frame motion correction followed by calculation of the contrast transfer function (CTF) parameters using CTFFIND4[48]. Micrographs at low estimated resolution were removed, resulting in 8786 micrographs for data processing. Using a template picker, 3,061,120 particles were extracted. After several rounds of 2D classification, 450,585 particles were subjected to ab initio reconstruction to produce an initial 3D model. After heterogeneous refinement, 202,972 particles of one class showing the IR$_{2xA62}$ complex were subjected to TOPAZ Train[49]. From the trained model, 1,484,860 particles were extracted. After several rounds of 2D classification, 1,311,269 particles were subjected to ab initio reconstruction and heterogeneous refinement. Duplicate particles were removed with a minimum separation distance of 100 Å. After removing duplicate

particles, 333,988 particles showing the complex were subjected to per-particle motion correction using local motion correction and global CTF refinement. Motion-corrected particles were subjected to additional ab initio and heterogeneous refinement. Finally, 181,797 particles yielded a map with a global resolution of 4.41 Å according to a Fourier shell correlation (FSC) criterion of 0.143. To improve the resolution of the ectodomain region, the final particles were subjected to local refinement with a mask covering the ectodomain. The final local refinement was performed with C2 symmetry, resulting in a resolution of 4.18 Å.

## Data processing for the IR$_{A62+Ins}$ complex

For the IR$_{A62+Ins}$ complex, dose-fractionated image stacks from 11,400 movies (dataset 1) were imported into CryoSPARC v3.3.1[47]. The imported images of dataset 1 were aligned and dose-weighted using full-frame motion correction, and CTF parameters were calculated using CTFFIND4[48]. Micrographs at low estimated resolution were removed, resulting in 10,929 micrographs for data processing. Using template picker, 2,890,222 particles from 10,929 micrographs were extracted. After several rounds of 2D classification, 35,547 particles were subjected to another round of particle sorting (particle set A). Using the trained TOPAZ model, 375,989 particles were extracted[49]. After 2D classification, 43,977 particles were subjected to another round of particle sorting (particle set B). Particle sets A and B were combined and subjected to heterogeneous refinement. After heterogeneous refinement, 25,921 particles showing the complex were subjected to an additional motion correction step for per-particle motion correction using local motion correction (particle set C). However, due to the small number of particles, particle set C could not produce a high-quality EM density map. To increase the number of particles, we collected an additional dataset.

The second dataset (12,923 movies) was aligned and dose-weighted using full-frame motion correction, and CTF parameters were calculated using CTFFIND4[48]. Micrographs at low estimated resolution were removed, resulting in 11,885 micrographs for data processing. Using template picker, 2,614,086 particles from 11,885 micrographs were extracted. After several rounds of 2D classification, 109,024 particles were subjected to ab initio reconstruction and heterogeneous refinement. A total of 59,135 particles of one class (particle set D) showing the IR$_{A62+Ins}$ complex were subjected to TOPAZ Train[49]. From the trained model, 915,328 particles were extracted. After several rounds of 2D classification, 742,426 particles were subjected to ab initio reconstruction and heterogeneous refinement. After heterogeneous refinement, 157,045 particles showing the complex were subjected to local motion correction (particle set E).

We combined the three datasets (C from dataset 1 and D, E from dataset 2) and removed duplicate particles with a minimum separation distance of 100 Å. A total of 244,774 particles were retained for subsequent 3D classification. After ab initio and heterogeneous refinement, 163,150 particles yielded a map with a global resolution of 4.27 Å according to an FSC criterion of 0.143. To improve the resolution of the aptamer density, the final particles were subjected to local refinement with a mask covering L1, CR, L2, FnIII-1, and A62. After additional CTF refinement using the improved map, the final local refinement resulted in a resolution of 3.95 Å.

## Data processing for the IR$_{A43+Ins}$ complex

For the IR$_{A43+Ins}$ complex, dose-fractionated image stacks from 10,836 movies were imported into CryoSPARC v3.3.1[47]. The imported images were subjected to beam-induced alignment and dose-weighting using full-frame motion correction followed by calculation of the CTF parameters using CTFFIND4[48]. Particles were automatically picked based on a template generated from the EM map of the IGF1R-IGF-1 complex (EMD-20524) in CryoSPARC v3.3.1[47]. After

1,083,091 particles were extracted from 10,583 micrographs, several rounds of 2D classification were performed to exclude bad particles. The selected 146,603 particles were used to pick particles using TOPAZ[49]. After particle extraction and 2D classification, the selected 1,605,119 particles were combined with the 146,603 template-based particles mentioned above. Duplicate particles were removed with a minimum separation distance of 100 Å. After several rounds of 2D classification, 1,056,472 particles were subjected to two rounds of 3D ab initio reconstruction and heterogeneous refinement. A total of 160,070 particles from conformationally homogeneous classes were subjected to non-uniform refinement and local motion correction, followed by global and local CTF refinement. Subsequently, the resulting particles were subjected to 3D ab initio reconstruction and heterogeneous refinement. The final subset of 156,334 particles, excluding poorly defined classes, was subjected to global non-uniform refinement, yielding a map with a global resolution of 3.7 Å, according to the FSC using the 0.143 cut-off. The resulting map was subjected to local non-uniform refinement using a mask on the ectodomain, resulting in a 3.62 Å map.

After the second Ab initio/Hetero refinement, we observed a subpopulation of particles (23%) with a T-shape. A total of 154,441 particles with T-shaped conformation was subjected to 3D ab initio reconstruction and heterogeneous refinement. In the next round, after excluding poorly defined particles, 130,844 particles were subjected to local motion correction, additional 3D ab initio reconstruction, and heterogeneous refinement. The final subset of 127,787 particles was subjected to global non-uniform refinement with C2 symmetry, yielding a map with a global resolution of 4.18 Å, according to the FSC using the 0.143 cut-off.

### Model building

Atomic model building of the $IR_{A43+Ins}$ complex was initiated by docking with the leucine-zippered human IR ectodomain (PDB: 6HN5, 6HN4)[23] into the 3.6 Å EM map using UCSF Chimera v1.15[50]. The model was manually adjusted in Coot[51]. The αCT helix, which does not participate in insulin binding, was positioned by overlaying the IGF1R-IGF-1 complex structure (PDB: 6PYH)[24] and changing the sequences to IR. A model of the A43 aptamer was built de novo from the EM map. The model was subjected to real-space refinement using PHENIX 1.14 with rigid body and secondary structure restraints[52]. The validated model has a MolProbity score of 1.81 and a clash score of 5.48[53].

The $IR_{A62+Ins}$ complex structures were built by docking the $IR_{A43+Ins}$ structure for the insulin-bound monomer and a crystal structure (PDB: 4ZXB)[33] for the A62-bound monomer. The A62 aptamer model was built de novo from the cryo-EM maps, which were sharpened using a locally refined map. There was no clear density for the αCT helix, which does not participate in insulin binding. The models were manually adjusted in Coot[51]. The upper parts and overall models were subjected to real-space refinement using PHENIX 1.14 with rigid body and secondary structure restraints[52]. The refined models have a MolProbity score of 1.74 and 1.82 and a clash score of 4.50 and 5.39 for distant and short forms, respectively[53].

The $IR_{2xA62}$ model was built by docking the $IR_{A62+Ins}$ complex model into the EM maps and manually adjusting in Coot[51]. There was no clear density for both αCT helices. The model was subjected to real-space refinement using PHENIX 1.14 with rigid body and secondary structure restraints[52]. The refined model has a MolProbity score of 1.91 and a clash score of 7.50[53].

The $IR_{2xIns}$ structure was built by docking the four insulin-bound IR model (PDB: 6PXV)[27] into the EM map using UCSF Chimera v1.15[50] and manually adjusting in Coot[51]. The model was subjected to real-space refinement using PHENIX 1.14 with rigid body and secondary structure restraints[52]. The refined model has a MolProbity score of 1.88 and a clash score of 6.07[53].

### Reagents and antibodies

Aptamers were synthesized by Aptamer Science, Inc. (Seongnam, Korea). Recombinant human Insulin (91077 C) were purchased from Sigma-Aldrich (St. Louis, MI, USA). Anti-insulin receptor β-subunit (sc-57342) antibody anti-phospho-insulin receptor (Y1150; sc-81500) were purchased from Santa Cruz Biotechnology (Santa Cruz, CA, USA). Anti-phospho-insulin receptor (Y1146; 80732) was purchased from Cell Signalling Technology (Danvers, MA, USA). Anti-phospho-insulin receptor (Y1322; 44-809 G), anti-phospho-insulin receptor (Y1316; 44-807 G), anti-phospho-insulin receptor (Y1150/Y1151; 44-804 G), anti-phospho-insulin receptor (Y960; 44-800 G) antibodies, goat anti-rabbit IgG (SA5-35571) and goat anti-mouse IgG (SA5-35521) secondary antibodies conjugated to DyLight 800 were purchased from Invitrogen (Carlsbad, CA, USA). Goat anti-rabbit IgG (926-68021) and anti-mouse IgG (926-68020) conjugated to IRdye 680LT were purchased from LI-COR (Lincoln, NE, USA). For western blotting, the anti-phospho- insulin receptor (Y1150) antibody was used at a 1:200 dilution, and other primary antibodies were used at a 1:1000 dilution. The secondary antibodies were used at a 1:20,000 dilution.

### Cell-based IR phosphorylation assay

CHO-K1 cells were maintained in Ham's F-12K medium (Welgene, Gyeongsan, Korea); Rat-1/hIR cells were maintained in high-glucose Dulbecco's modified Eagle's medium (DMEM) with 10% (v/v) fetal bovine serum (FBS; Gibco) and antibiotic-antimycotic (Gibco). All cells were incubated at 37 °C under a humidified atmosphere containing 5% $CO_2$ prior to experiments. One day prior to transfection, cells were seeded in 12-well plates with Opti-MEM medium (Gibco) containing 1% FBS. Transfection of WT or mutant IRs was performed with Lipofectamine 3000 (Invitrogen, Carlsbad, CA, USA) according to the manufacturer's instructions. After 24 h, the medium was replaced with Ham's F-12K medium containing 10% FBS. Cells were incubated for an additional 48 h before being used in experiments. Aptamers and insulin were prepared in Krebs–Ringer HEPES buffer (25 mM HEPES pH 7.4) containing 120 mM NaCl, 5 mM KCl, 1.2 mM $MgSO_4$, 1.3 mM $CaCl_2$, and 1.3 mM $KH_2PO_4$. All aptamer samples were heated for 5 min at 95 °C and slowly cooled to room temperature to reconstitute the tertiary structure of the aptamer. Before insulin or aptamer stimulation, cells were serum-starved for 3 h, then treated with insulin or IR-A62 aptamer at the indicated concentrations. After stimulation of cells with insulin or aptamer for the described time, cells were washed with cold phosphate-buffered saline (PBS) and lysed in cell lysis buffer (50 mM Tris-HCl pH 7.4) containing 150 mM NaCl, 1 mM EDTA, 20 mM NaF, 10 mM glycerophosphate, 2 mM $Na_3VO_4$, 1 mM phenylmethylsulfonyl fluoride (PMSF), 10% glycerol, 1% Triton-X, 0.1% SDS, 0.5% sodium deoxycholate, and protease inhibitor cocktail. Cell lysates were isolated by centrifugation at 14,000 rpm for 15 min at 4 °C, and the supernatant was mixed with 5× Laemmli sample buffer. Proteins in cell lysates were separated by SDS-PAGE, transferred to a nitrocellulose membrane, blocked with 5% skim milk for 30 min, then probed with primary antibody at 4 °C overnight. Blotting was performed using Odyssey infrared imaging system (LI-COR, Lincoln, NE, USA). Detailed information on reagents is provided in Supplementary Table 2.

### Reporting summary

Further information on research design is available in the Nature Research Reporting Summary linked to this article.

## Data availability

The data that support this study are available from the corresponding authors upon reasonable request. The cryo-EM density maps have been deposited in the Electron Microscopy Data Bank (EMDB) under accession codes EMD-34021 ($IR_{2xA62}$), EMD-34020 (overall refined $IR_{A62+Ins}$), EMD-34019 (locally refined $IR_{A62+Ins}$), EMD-34018 ($IR_{A43+Ins}$), and EMD-34281 ($IR_{2xIns}$). The coordinates have been in the RCSB

Protein Data Bank (PDB) under accession codes 7YQ6 (IR$_{2xA62}$), 7YQ5 (overall refined IR$_{A62+Ins}$), 7YQ4 (locally refined IR$_{A62+Ins}$), 7YQ3 (IR$_{A43+Ins}$), and 8GUY (IR$_{2xIns}$). Source data are provided with this paper.

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

## Acknowledgements

This work was supported by grants from the National Research Foundation of Korea (NRF) funded by the Korean government (MEST, No. 2021R1A2C301335711, 2017M3A9F6029736, and 2019M3E5D6066058 to Y.C., 2016K1A1A2912722 to S.H.R., and 2020M3H1A1075314 to N.-O.Y.) and the BK21 program (Ministry of Education) to Y.C. and S.H.R.

## Author contributions

J.K. carried out protein expression, purification, and structure determination with the help of Y.K. and J. K.; N.-O.Y. performed biochemical experiments with the help of M.P., S.P., and J.N.; J.K., N.-O.Y., S.H.R., and Y.C. designed the research; N.-O.Y., S.H.R., and Y.C. wrote the manuscript.

## Competing interests

The authors declare no competing interests.
