## [Peer Review File · Nature Communications]

Functional selectivity of insulin receptor revealed by aptamer-trapped receptor structuresReviewers' Comments:

Reviewer #1:

Remarks to the Author:

The manuscript by Kim et al. reports structural (cryo-EM) and functional data on two DNA aptamers, one of which (A62) acts as a biased agonist (in the absence of insulin) for the homodimeric insulin receptor (IR), and the other of which (A43) acts as a positive allosteric modulator (in the presence of insulin) of the IR.

Three cryo-EM structures are presented – IR:2A62, IR:insulin:A62, and IR:insulin:A43 – with overall resolutions of 3.6 to 4.5 Å. In general, the structures recapitulate several of the various conformational states of the IR ectodomain observed in previous cryo-EM and crystal structures, with the aptamers binding to one or both of the distinct insulin binding sites, S1 and S2 (or their symmetric counterparts, S1' and S2').

Importantly, the authors monitor tyrosine phosphorylation of the beta subunit in response to stimulation by either insulin or the aptamers (+/- insulin). They find that insulin stimulation results in full phosphorylation of the beta subunit: the three activation-loop tyrosines (Y1146/1150/1151), the juxtamembrane tyrosine (Y960), and the C-terminal tyrosines (Y1316/1322), whereas saturating amounts of A62 result in predominantly Y1150 phosphorylation (hence, biased agonist).

Based on the structural data, the authors surmise that full phosphorylation of the IR requires close apposition (~26 Å) of the FnIII-3 domains (which lead into the transmembrane (TM) helices), as viewed in the IR:insulin:A43 structure, versus the 34-45 Å separation observed in the IR:2A62 and IR:insulin:A62 structures. To test this hypothesis, they introduced a 10-residue flexible linker between the end of FnIII-3 and the TM helix and found that stimulation by insulin now resulted in biased phosphorylation: phosphorylation of Y1150 but not of the other beta subunit sites. Mutation of a tyrosine (Y972) in the juxtamembrane region, which is known to act as a negative regulator of IR trans-autophosphorylation, resulted in full phosphorylation of the IR when stimulated with A62.

Conclusions:

These structural and functional data on the IR are novel and potentially highly insightful, particularly the linkage between FnIII-3 separation and beta-subunit phosphorylation. The model as presented in Fig. 6 indicates that Y1150 phosphorylation can occur at an intermediate stage of receptor conformational change induced by insulin, and that full phosphorylation of the beta subunit requires the close apposition of the two FnIII-3 domains along with conformational changes in the cytosolic juxtamembrane region. In some ways, the data and model lead to more questions than answers, particularly as they pertain to the step-wise phosphorylation/activation process.

Points:

1) Although we know that Y1150 is unique in the IR activation loop because it acts as a pseudosubstrate inhibitor and is the first tyrosine phosphorylated during activation, it's not clear from the data or model why phosphorylation of this tyrosine is less sensitive to kinase proximity than Y1151 or Y1146 in the activation loop, unless Y1150 is phosphorylated in cis rather than in trans, as for the rest of the sites in the activation loop and elsewhere. However, there are no reliable data supporting cis phosphorylation of Y1150. The authors might wish to comment on this issue.

In general, what is happening in the cytoplasmic domains upon insulin binding and during the phosphorylation process is intriguing and important, and the current study provides some hints as to what might be going on mechanistically.

2) In lines 230-231, "These results indicate that the spatial proximity of both kinases is critical for only m-pY1150, and normal..." The first half of this sentence is confusing, because in the model presented in Fig. 6 and discussed in the text, it is the close apposition of the two FnIII-3 domains (~26 Å) that

allows phosphorylation of the remainder of the phosphorylation sites in the beta subunit, and that Y1150 phosphorylation can occur with a larger spatial separation.

Minor points:

1) In Fig. 5C, right cartoon panel, Y972 is shown upstream of pY960 (red circle) instead of downstream.

2) In Ext. Data Fig. 10, insulin is misspelled ("insuin") numerous times.

Reviewer #2:

Remarks to the Author:

In the current paper the authors (Kim and colleagues) report cryoEM structures of the insulin receptor (IR), in complex with two different aptamers that modulate IR structure and function. These aptamers have been described previously, so what is new is the structures, which no doubt have value. The authors then use the structures to propose a model of IR activation, which in my opinion also has value, but cannot be considered supported by data. For instance, the authors find that A62 binds in a novel distinct way to IR extracellular domain. Then they propose that this structure is a biological relevant intermediate in the activation after insulin binds to IR. This is possible, of course, but not necessarily true. The model that the authors propose in Figure 6 is a kinetic model, consisting of consecutive steps of activation. Such a model cannot be justified by the data in the paper, which are not acquired over time.

There are three new structures. One is of the aptamer A62 bound to IR in the absence of ligand. The second one is of A62 bound to IR in the presence of insulin. The third one is the structure of the aptamer A43 bound to IR in the presence of insulin. The extracellular domains in all these structures are well resolved, providing details as to how exactly the aptamers bind and what types of conformation of the extracellular domains get stabilized. This structural information is highly valuable, but I feel that the authors over-interpret the results.

One insight that comes from the structures is that the distance between the membrane-proximal domains are likely important for activation. This confirms prior work which has shown that the proximity of the TM domains is important for activation of IGF-1R. I think this prior work should be discussed, as well as the fact that the authors cannot say anything about the intracellular domains as they don't measure distances between the intracellular domains (they only have extracellular domain structures and phosphorylation measurements). Despite this, on page 8 the authors talk about "spatial proximity of both kinases". But this is pure speculation which I think should be removed.

To sum up, what the authors are proposing is a model that cannot be verified until structural information of some kind is obtained about the kinase domains. The authors cannot resolve the kinase domains or the TM domains, so they do not know how the kinases are responding to the extracellular domain conformations. One imagines that completely new mechanisms will be revealed once the kinase domains are resolved.

Further, I am somewhat confused as to how the authors define and interpret bias. Bias has to do with the differential engagement of different distinct signaling cascades downstream of a receptor. The phosphorylation of a tyrosine in the activation loop of the kinase is required for the activation of the kinases and the phosphorylation of all other tyrosines, so the two measured responses (activation loop phosphorylation and other Y phosphorylation) are not independent. The authors' thinking is different from the accepted views in pharmacology, detailed by Kenakin and many other authors. The authors are claiming on page 10 that the new structures provide mechanistic insights into RTK bias, but I cannot see what these insights are.

I also think the paper needs to be rewritten prior to publication in any journal. Currently it is written for people who have worked all their lives with the IR. It is full of abbreviations that are not defined. The average reader will wonder what "alphaCT" is. "Displacement of alphaCT"? From where? There are thousands of researchers working on membrane receptors, and the authors must make this text accessible to all of them. The introduction of the aptamers should be also for written with this general readership in mind.

Responses to reviewers' comments

Responses to reviewer #1's comments:

Points:

Q1. Although we know that Y1150 is unique in the IR activation loop because it acts as a pseudosubstrate inhibitor and is the first tyrosine phosphorylated during activation, it's not clear from the data or model **why phosphorylation of this tyrosine is less sensitive to kinase proximity than Y1151 or Y1146 in the activation loop**, unless Y1150 is phosphorylated in *cis* rather than in *trans*, as for the rest of the sites in the activation loop and elsewhere. However, there are no reliable data supporting *cis* phosphorylation of Y1150. **The authors might wish to comment on this issue.**

>> **Please see the revised text line 320 to 338 in P11.** We also included two experimental data (see below), which we did not include in the manuscript (but we can add them depending on the reviewer's suggestion). We agree with a reviewer's concern. Although many studies support that IR autophosphorylation occurs in *trans*, we also considered the possibility of *cis*-phosphorylation of Y1150. To resolve this issue, **we performed the following experiment.**

We introduced a kinase-dead (KD, K1018A) mutation into YFP-tagged IR to allow *trans*-autophosphorylation to occur in only one direction and, transfected a 1:2 mixture of YFP-tagged KD-IR (upper band) and untagged WT-IR (Lower band) to distinguish the phosphorylation state of each protomer (please see Fig. a in the next page). In the WT+KD/YFP hybrid receptor, the WT kinase can phosphorylate the KD kinase in *trans* or can be phosphorylated in *cis*, but cannot be phosphorylated by the KD kinase in *trans*. After insulin stimulation, we observed m-Y1150 is induced in the KD kinase. Moreover, we performed immunoprecipitation of YFP to observe the phosphorylation in WT kinases in the hybrid receptor. However, phosphorylation was not observed in the WT kinase in the hybrid receptor. This result suggests that m-pY1150 occurs only in *trans* in IR dimer.

Alternatively, we considered the possibility of differences in substrate specificity of Y1146, Y1150 and Y1151. That is because Y1150 is an efficient substrate of IR kinase, m-Y1150 is simply regulated by the spatial proximity of both kinases in IR dimer. However, because Y1146 and Y1151 are not as efficient as Y1150 for IR kinase, the phosphorylation of Y1146 and Y1151 may require some conformational rearrangement of intracellular domain

to overcome peptide specificity of IR kinase. We have fully discussed this in the text.

We also examined whether the change in peptide specificity of the IR kinase was dependent on phosphorylation status by performing an *in vitro* kinase assay using partially purified IR (Fig. b). Basal IR (non-phosphorylated) following serum starvation and activated IR (phosphorylated) following insulin stimulation were immunoprecipitated from Rat-1 cells or Rat-1 cells stably overexpressing IR (Rat-1/IR). Each purified IR was incubated with a substrate peptide containing the IR activation loop sequence (A-loop) in the presence of ATP (Fig. c). Basal IR kinase induced m-pY1150, but the level of m-pY1150 induced by activated IR kinase was higher than basal IR kinase (Fig. d). However, pY1150/pY1151 and pY1146 on A-loop peptides were not detected in either IR kinase (Fig. e, f). Because dual phosphorylation of a substrate requires that it contacts the kinase twice, the probability of dual phosphorylation is less than mono-phosphorylation. Thus, we also performed an *in vitro* kinase assay using a substrate containing m-pY1150 (A-loop(pY1150)), but pY1150/pY1151 was still not detected (Fig. e). These results indicate that, of the **three tyrosine residues of the activation loop, only Y1150 is effectively recognized by IR kinase as a substrate**. Moreover, phosphorylation of the activation loop increases the activity of IR kinase, but does not change the peptide specificity of IR kinase.

a. To generate hybrid IRs, CHO-K1 cells were transiently cotransfected with untagged-IR (lower bands) and YFP-tagged IR (upper bands). After serum starvation for 1 hour, the cells were stimulated with 200 nM insulin for 10 min. **WT**=wild type, **YFP**=yellow fluorescent protein, **KD**=kinase-dead mutation. **b.** Cartoon representation of *in vitro* IR kinase assay with peptide substrates. **c.** Sequence information on peptide substrates used for *in vitro* IR kinase assay. **d, e, f.** IR was partially purified from Rat-1 or Rat-1/hIR cells by immunoprecipitation. Purified IRs were incubated with biotin-conjugated substrate peptides for 60 min at 36°C. After peptide immobilization on streptavidin-coated 96-well plates, phosphorylation was detected using site-specific primary antibody and alkaline phosphatase-conjugated secondary antibody.

Q2. In general, what is happening in the cytoplasmic domains upon insulin binding and during the phosphorylation process is intriguing and important, and **the current study provides some hints as to what might be going on mechanistically.**

In lines 230-231, “These results indicate that the spatial proximity of both kinases is critical for only m-pY1150, and normal...” **The first half of this sentence is confusing**, because in the model presented in Fig. 6 and discussed in the text, it is the close apposition of the two FnIII-3 domains (~26 Å) that allows phosphorylation of the remainder of the phosphorylation sites in the beta subunit, and that Y1150 phosphorylation can occur with a larger spatial separation.

>> Please see P10, line 291 to 294. Our original intention was to emphasize that some conformational rearrangement of intracellular domains is essential for multi-phosphorylation of IR. We have revised this sentence: “These results suggest that the spatial proximity of FnIII-3 membrane-proximal ends is important for m-pY1150, but multi-phosphorylation of IR requires additional conformational rearrangement of intracellular domains mediated by the tight coordination between TM domains and FnIII-3 ends.”

Minor points:

1) In Fig. 5C, right cartoon panel, Y972 is shown upstream of pY960 (red circle) instead of downstream.

>> We have revised Fig. 5C to correctly reflect order of two amino acid residues.

2) In Ext. Data Fig. 10, insulin is misspelled (“insuin”) numerous times.

>> We have removed Ext. Data Fig. 10, while retaining the related contents in the discussion section.

Responses to reviewer #2 (Remarks to the Author):

In addressing second reviewer's questions, we focused on the following points. First, we removed the possible overinterpretation of the A62-bound IR structures. We explain how the aptamer-bound IR conformations are responsible for the selective activation of IR. We also provide basis why the aptamer-bound IR conformations represent the plausible intermediate states. Second, we removed the detailed model for the intracellular kinase domains in the activation process. We fully revised Figure 6. Third, we removed the term "biased agonist". Instead we used "aptamer agonist" or "selective agonist". Fourth, we extensively revised the introduction and discussion sections, which contain the points that resolve the reviewer's concerns.

Q1. "...These aptamers have been described previously, so what is new is the structures, which no doubt have value. The authors then use the structures to propose a model of IR activation, which **in my opinion also has value, but cannot be considered supported by data**. For instance, the authors find that A62 binds in a novel distinct way to IR extracellular domain. Then they propose that **this structure is a biological relevant intermediate in the activation** after insulin binds to IR. This is possible, of course, but not necessarily true. The model that the authors propose in Figure 6 is a kinetic model, consisting of consecutive steps of activation. Such a model cannot be justified by the data in the paper, which are not acquired over time..... The extracellular domains in all these structures are well resolved, providing details as to how exactly the aptamers bind and what types of conformation of the extracellular domains get stabilized. **This structural information is highly valuable, but I feel that the authors over-interpret the results.**

>> Please see the first to third paragraphs (line 310 to 356 in P11 and P12) in the revised discussion. We carefully interpret the structure of the A62-bound IRs. We fully discussed the basis for the functional selectivity of the A62 bound IR conformations.

Q2. One insight that comes from the structures is that the distance between the membrane-proximal domains are likely important for activation. **This confirms prior work which has**

shown that the proximity of the TM domains is important for activation of IGF-1F. I think this prior work should be discussed, as well as the fact that the authors cannot say anything about the intracellular domains as they don't measure distances between the intracellular domains (they only have extracellular domain structures and phosphorylation measurements). Despite this, on page 8 the authors talk about “**spatial proximity of both kinases**”. But this is pure speculation which I think should be removed. ... The authors cannot resolve the kinase domains or the TM domains, so they do not know how the kinases are responding to the extracellular domain conformations. One imagines that completely new mechanisms will be revealed once the kinase domains are resolved.

>> Please see revised P9 (line 275 to 277) and P10 (line 288 to 294). We agree with the reviewer as we do not know the conformational changes of kinase domains at the molecular level, and thus we removed the speculation part of the kinase conformational change from the results and discussion (line 304 to 306 in P11). We fully discussed the importance of the translocation of the membrane proximal ends of FnIII-3 in mono-phosphorylation and selective metabolic signaling. We also discussed the conformational changes of IGF-1R induced by IGF1 binding.

Q3. Further, I am somewhat confused as to how the authors **define and interpret bias**. Bias has to do with the differential engagement of different distinct signaling cascades downstream of a receptor. The phosphorylation of a tyrosine in the activation loop of the kinase is required for the activation of the kinases and the phosphorylation of all other tyrosines, so the two measured responses (activation loop phosphorylation and other Y phosphorylation) are not independent. The authors' thinking is different from the accepted views in pharmacology, detailed by Kenakin and many other authors. **The authors are claiming on page 10 that the new structures provide mechanistic insights into RTK bias**, but I cannot see what these insights are.

>> We removed the term “biased agonism” throughout the text.

Q4. I also think the paper needs to be rewritten prior to publication in any journal. Currently it is written for people who have worked all their lives with the IR. It is full of abbreviations that are not defined. The average reader will wonder what “alphaCT” is. “Displacement of alphaCT”? From where? There are thousands of researchers working on membrane receptors,

and the authors must make this text accessible to all of them. The introduction of the aptamers should also be written with this general readership in mind.

>> We have extensively revised the introduction for general readers as suggested by the reviewer. We also added architecture of insulin receptor in supplementary Fig. 1a. Please see the revised text line 137 in page 5 and line 148 – 150 in page 6.

Response to the reviewer #1's comments:

In the revised manuscript, the authors have adequately addressed the concerns I raised previously, but additional (minor) concerns are now present.

1) Page 10, lines 310-313. This sentence, although revised, does not quite convey the correct idea. I suggest something like the following: "These results suggest that close (~26 Å) apposition of the FnIII domains are required for full (multi) phosphorylation of the cytoplasmic domains, but that monophosphorylation of Y1150 can occur at intermediate FnIII distances (34-45 Å)."

>> We thank to the reviewer for very helpful comment. Please see lines 320-324, P10. We have revised the text as suggested by the reviewer.

2) Page 10, line 303. "obviously" should be deleted.

>> We have deleted "obviously" from the sentence in line 302.

3) Page 11, lines 344 and onward. I would not even mention the possibility of cis phosphorylation, because it has been proven not to occur – previously and by the authors (data provided in the rebuttal letter). I would follow the sentence ending "but our results showed that this does not happen." with "A plausible explanation is that Y1150 is the most efficient..."

>> Please see the revised sentences in P11, lines 352-354. We have removed the sentence containing "cis-phosphorylation...". We also included a sentence "A plausible explanation is that Y1150 is..." in line 354.

4) Page 11, line 356. "Conformational pressure" is not a term that is used in the field. Perhaps it is as simple as this: the Y972 cis-autoinhibitory interaction with the kinase domain prevents facile trans-phosphorylation of all but Y1150 (which still requires insulin), and that maintaining the kinase domains in close proximity via the L1-FnIII-2/-2' interaction (~26 Å) is necessary to achieve full phosphorylation.

>> We very much appreciate the reviewer for this helpful suggestion. We have included this sentence in line 367-370 and removed the sentences containing "...conformational pressure..".

5) I would argue that Supplementary Videos 1 & 2 are not directly related to the structural data presented in the current study – they summarize previous structural data – and should be omitted.

>> We have removed the two Videos.

Reviewer #2 (Remarks to the Author):

The authors have made significant edits to the paper, such that the paper is now much improved. In particular, they have removed speculations about the behavior of the kinase domains which are not resolved, and they now focus the discussion on the extracellular domain structures only.

To me it is not clear whether the structures the authors solve are indeed intermediates in the activation of IR. I guess only the future can tell if these structures are biologically relevant.

>> We thank to a reviewer to guide us to remove speculations and focus on discussion on the ectodomain.

Response to the reviewer #3's comments

Comments:

1. "... I suggest mentioning in the first paragraphs (e.g., around line 120) that only the ECD region of the receptor was determined."

>> Please see line 120. We have included "...IR ectodomain...". We also note that "While all ectodomains are well ordered, the TM and intracellular domains are not visible." is included in lines 142-143.

2. "... I suggest that the resolution is reported with the same number of decimal places (ideally 2) in both the Supplementary Table and the main text."

>> Please see lines 121, 140, 219 and Supplementary Table. We have reported the resolution with two decimal places.

3. Line 210 states, "Consequently, the distance between L1 and FnIII-1' in IR A62+Ins is closer than the site-1/site-2' insulin coordination in the reported structures (Supplementary Fig. 4f-h)."

I suggest that these measured distances be indicated in the figure.

>> Please see the revised Supple Fig 4f-h. We have included measured distances between L1 and FnIII-1'.

4. The insulin binding sites-1, -1', -2, -2' are not clearly labeled in the figures. I suggest labeling the binding sites in at least one of the figures of the IR -complex. This should help readers understand where the binding sites are located.

>> Please see the revised Figs 1f, 2a, b, Supple Fig 4c

Questions:

1. The dataset from IR A62+Ins is split into two parts and processed separately. Is there a specific reason for processing the data in this way? I suggest adding a description to the methods section to explain why the data was processed in this

particular way.

>> We thank a reviewer for this comment. In describing the process of IR A62+Ins data set, we did not describe it clearly and we revised this part in the revised text. Briefly, we collected two datasets independently and processed it. After initial processing with first dataset, we realized that the number of particles is not complete enough to obtain high quality map, so we collected additional dataset. We described this process in full in the revised method section. Line 642-645 (P20, 21), lines 670-697 (P21, 22)

2. In Supplementary Figure 5a, after the second Ab-initio/Hetero refinement, a conformation is shown in pink that is populated at 23%. This conformation looks like a T-shape. Did you analyze this class further? What were you able to see for this subpopulation of the receptor?

>> The reviewer is correct. Indeed, we observed a T-shaped conformation. In this structure, two insulin molecules symmetrically bind to the IR, similar to that reported structures. We omitted this in the previous text, but now included with a figure. We appreciate a reviewer's comment. Lines 243-249 (P8), Line 726-733 (P23), line 757-760 (P24) and supplementary figure 8. The PDB and map have been deposited (line 566, P18)

Response to the reviewer #1's comments:

In the revised manuscript, the authors have adequately addressed the concerns I raised previously, but additional (minor) concerns are now present.

1) Page 10, lines 310-313. This sentence, although revised, does not quite convey the correct idea. I suggest something like the following: "These results suggest that close (~26 Å) apposition of the FnIII domains are required for full (multi) phosphorylation of the cytoplasmic domains, but that monophosphorylation of Y1150 can occur at intermediate FnIII distances (34-45 Å)."

>> We thank to the reviewer for very helpful comment. Please see lines 320-324, P10. We have revised the text as suggested by the reviewer.

2) Page 10, line 303. "obviously" should be deleted.

>> We have deleted "obviously" from the sentence in line 302.

3) Page 11, lines 344 and onward. I would not even mention the possibility of cis phosphorylation, because it has been proven not to occur – previously and by the authors (data provided in the rebuttal letter). I would follow the sentence ending "but our results showed that this does not happen." with "A plausible explanation is that Y1150 is the most efficient..."

>> Please see the revised sentences in P11, lines 352-354. We have removed the sentence containing "cis-phosphorylation...". We also included a sentence "A plausible explanation is that Y1150 is..." in line 354.

4) Page 11, line 356. "Conformational pressure" is not a term that is used in the field. Perhaps it is as simple as this: the Y972 cis-autoinhibitory interaction with the kinase domain prevents facile trans-phosphorylation of all but Y1150 (which still requires insulin), and that maintaining the kinase domains in close proximity via the L1-FnIII-2/-2' interaction (~26 Å) is necessary to achieve full phosphorylation.

>> We very much appreciate the reviewer for this helpful suggestion. We have included this sentence in line 367-370 and removed the sentences containing "...conformational pressure..".

5) I would argue that Supplementary Videos 1 & 2 are not directly related to the structural data presented in the current study – they summarize previous structural data – and should be omitted.

>> We have removed the two Videos.

Reviewer #2 (Remarks to the Author):

The authors have made significant edits to the paper, such that the paper is now much improved. In particular, they have removed speculations about the behavior of the kinase domains which are not resolved, and they now focus the discussion on the extracellular domain structures only.

To me it is not clear whether the structures the authors solve are indeed intermediates in the activation of IR. I guess only the future can tell if these structures are biologically relevant.

>> We thank to a reviewer to guide us to remove speculations and focus on discussion on the ectodomain.

Response to the reviewer #3's comments

Comments:

1. "... I suggest mentioning in the first paragraphs (e.g., around line 120) that only the ECD region of the receptor was determined."

>> Please see line 120. We have included "...IR ectodomain...". We also note that "While all ectodomains are well ordered, the TM and intracellular domains are not visible." is included in lines 142-143.

2. "... I suggest that the resolution is reported with the same number of decimal places (ideally 2) in both the Supplementary Table and the main text."

>> Please see lines 121, 140, 219 and Supplementary Table. We have reported the resolution with two decimal places.

3. Line 210 states, "Consequently, the distance between L1 and FnIII-1' in IR A62+Ins is closer than the site-1/site-2' insulin coordination in the reported structures (Supplementary Fig. 4f-h)."

I suggest that these measured distances be indicated in the figure.

>> Please see the revised Supple Fig 4f-h. We have included measured distances between L1 and FnIII-1'.

4. The insulin binding sites-1, -1', -2, -2' are not clearly labeled in the figures. I suggest labeling the binding sites in at least one of the figures of the IR -complex. This should help readers understand where the binding sites are located.

>> Please see the revised Figs 1f, 2a, b, Supple Fig 4c

Questions:

1. The dataset from IR A62+Ins is split into two parts and processed separately. Is there a specific reason for processing the data in this way? I suggest adding a description to the methods section to explain why the data was processed in this

particular way.

>> We thank a reviewer for this comment. In describing the process of IR A62+Ins data set, we did not describe it clearly and we revised this part in the revised text. Briefly, we collected two datasets independently and processed it. After initial processing with first dataset, we realized that the number of particles is not complete enough to obtain high quality map, so we collected additional dataset. We described this process in full in the revised method section. Line 642-645 (P20, 21), lines 670-697 (P21, 22)

2. In Supplementary Figure 5a, after the second Ab-initio/Hetero refinement, a conformation is shown in pink that is populated at 23%. This conformation looks like a T-shape. Did you analyze this class further? What were you able to see for this subpopulation of the receptor?

>> The reviewer is correct. Indeed, we observed a T-shaped conformation. In this structure, two insulin molecules symmetrically bind to the IR, similar to that reported structures. We omitted this in the previous text, but now included with a figure. We appreciate a reviewer's comment. Lines 243-249 (P8), Line 726-733 (P23), line 757-760 (P24) and supplementary figure 8. The PDB and map have been deposited (line 566, P18)

Reviewers' Comments:

Reviewer #3:

Remarks to the Author:

The questions I posed were addressed in this revised manuscript.

Structures determined in this study are valuable in the community and they will contribute to future studies of Insulin receptor.